# Exploring the Consumers' Purchase Intention on Online Community Group Buying Platform during Pandemic

**Mengyao Zhang** [1,2] , **Hasliza Hassan** [1,*] **and Melissa Wendy Migin** [1]

1  Faculty of Management, Multimedia University, Cyberjaya 63100, Malaysia
2  School of Business Administration, Shandong Women's University, Jinan 250300, China
*  Correspondence: hasliza.hassan@mmu.edu.my

**Abstract:** One of the main methods of shopping for many consumers during the COVID-19 pandemic was through online community group-buying. This shopping method caters to the consumer demand of reducing contact and centralized procurement. However, some online community group-buying platforms could not attract many consumers, and consumer participation was low. Therefore, determining which factors affect consumers' willingness to use online community group buying is important. Based on the Unified Theory of Acceptance and Use of Technology (UTAUT) and perceived risk theory, this research explores the effects of performance expectancy, effort expectancy, social influence, facilitating conditions, and perceived risk on consumers' willingness to use online community group buying. In this research, a questionnaire survey was used, and the sample randomly collected from online consumers who had experience in online community group buying. A total of 280 respondents were collected. The collected data were analyzed by descriptive statistics, reliability, validity, correlation, and regression analysis. The results show that performance expectancy, effort expectancy, and social influence have a significant positive effect on the purchase intention of community group-buying consumers, while facilitating conditions and perceived risk have no significant positive effect. This research further enriched and improved the research on the use intention of an online community group-buying platform by integrating the UTAUT model and perceived risk theory. In practice, this research provides a new perspective and practical reference for how the online community group-buying platform can better attract consumers and maintain sustainable long-term customer relations.

**Keywords:** online community group buying; UTAUT model; perceived risk

## 1. Introduction

With the rapid development of digital technology, online shopping has become an important consumption habit for many people. China has 1051 million Internet users as of June 2022, with a 744.4% Internet penetration rate. The number of online shopping users in China is 841 million, accounting for 80% of the total Internet users. The network's popularity is the basis for the development of the online community group buying (OCGB). The further popularity of online payment has promoted the continuous expansion of the scale of OCGB platform users. The group buying model started in 2008 on the Groupon website in the United States [1]. Before the pandemic in 2016, China's online OCGB platform developed rapidly. Compared with traditional e-commerce, community group purchase can be delivered directly from the origin or transit warehouse, eliminating intermediate links, and has price and timeliness advantages. Consumers can browse the commodities on the online community group purchase platform and place an order to complete the group purchase. After the group purchase is successful, the community group purchase does not need to deliver the goods one by one to the addresses of the customers. Instead, consumers can pick up the goods at the designated offline stores at a preferential price [2,3].

In 2018, the number of Chinese OCGB platform consumers exceeded 300 million, and the market size of the OCGB industry reached 7.36 billion. More than 30 OCGB platforms emerged, including Prosperity Selected, Meituan Selected, and Squirrel Pinpin. Since COVID-19 broke out between 2020 and 2022, consumers have exhibited a preference for reduced number of trips and purchase products with as little personal exposure to others as possible. The contactless distribution mode of OCGB has certain advantages that, greatly meet the needs of people who find travel to be inconvenient. OCGB is well known and is being used increasingly. By the end of 2021, 285 OCGB platforms are expected to be established in China and the market size of OCGB in China will reach 122 billion yuan in 2022.

However, the exceedingly high number of OCGB enterprises competing on the same platform has given rise to seriousproduct homogeneity and fierce competition. Consumers in the same region often use three or four OCGB platforms [4]. Therefore, how consumers can be attracted to choose a specific OCGB platform and increase consumers' willingness and loyalty to use OCGB are issues facing most OCGBs. Several studies on the use intention of consumers in OCGB have been conducted, with most of these studies focusing on online shopping [5]. Therefore, based on the Unified Theory of Acceptance and Use of Technology (UTAUT) model with high explanatory power for consumer adoption intention, combined with the theory of perceived risk, this research explores the influencing factors of OCGB consumers' online purchase intention and develops a theoretical model of the factors that influence users' intention to utilize OCGB platforms. The empirical findings of this research are expected to provide inspiration and suggestions for improving the consumer experience of the OCGB platform, enhancing the brand trust and preference of consumer OCGB platform, attracting more consumers to use the OCGB platform and providing suggestions for the operation and sustainable development of the OCGB platform.

## 2. Literature Review and Theoretical Foundation

Online community group buying refers to a real community based offline, with community leaders (usually store operators around the community) as distribution nodes, using WeChat groups, applications, and other mobile platforms to collect the needs of people in the same community. Consumers go to the community to purchase goods through online payment. Li et al. (2020) [6] believed that OCGB was a new group-buying mode similar to "online ordering" and "community picking up", which was completed by group-buying leaders relying on physical channels. OCGB is a derivative of traditional online group buying that integrates the characteristics of online group buying and offline community shopping [7]. Limited research on the OCGB platform has been conducted, and most studies have focused on online group buying and social e-commerce. Few studies have been conducted from the perspective of consumers. For example, Hsu et al. (2014) [8] analyzed the reasons for the formation of consumer willingness and behavior patterns in the online shopping process from the perspective of e-commerce and confirmed that consumer willingness is the main factor affecting online shopping behavior. Cheng and Huang (2013) [9] further found that the reputation, product quality, number of coupons, number of re-purchase consumers, and other factors of online group buying affect online shopping behavior. Qing et al. (2018) [10] pointed out that the seller's reaction to online consumer negative comments (NCRs) has a significant effect on consumers' purchase intentions. Ni et al. (2019) [11] found that price discounts and services in OCGB affect consumers' purchase intentions. Chen et al. (2011) [12] studied the influence of trust on online group buying and determined that consumers have different intentions to participate in group-buying activities under different trust scenarios.

This research is based on the UTAUT model. Venkatesh et al. (2003) [13] proved that this model has a strong explanatory power for consumers' willingness to accept new technologies in the organizational context. The UTAUT model refines the factors that affect consumer behavior intentions into four core independent variables, namely, performance expectancy (PE), effort expectancy (EE), social influence (SI), and facilitating conditions

(FC). Scholars often use the UTAUT model in studying the user intention and behavior of various information technology systems. For example, Celik (2016) [14] used the UTAUT model to explore the effects of anxiety on customers' adoption of online shopping. Nur and Panggabean (2021) [15] used the extended UTAUT model to analyze the factors affecting the adoption of mobile payment as a payment method in Generation Z. Chen et al. (2021) [16] used the UTAUT model to explore the determinants of purchase intention when using the fresh e-commerce platform.

Bauer (1960) [17] extended the concept of perceived risk from psychology, which was later adopted in other disciplines. Bauer studied consumers' purchase behavior and found that the consequences of purchase behavior are not always satisfactory and may also have purchase consequences that make consumers feel unhappy. Consumers cannot accurately judge what kind of situation will occur. Such consumers have uncertainty that their behavior may lead to adverse consequences before the behavior starts, which is called perceived risk. Perceived risk is used more when studying consumers' wishes and behaviors. For example, Chang et al. (2016) [18] studied online shopping behavior and found that perceived risk negatively affects purchase intention. Li et al. (2020) [6] studied consumers' online furniture purchase behavior and found that perceived risk significantly negatively affects consumers' intent to make online purchases.

## 3. Research Model and Hypotheses

Based on previous research, this research proposes PE, EE, SI, perceived risk and facilitating conditions drive purchase intention based on the UTAUT model and perceived risk theory. Figure 1 shows the proposed hypotheses.

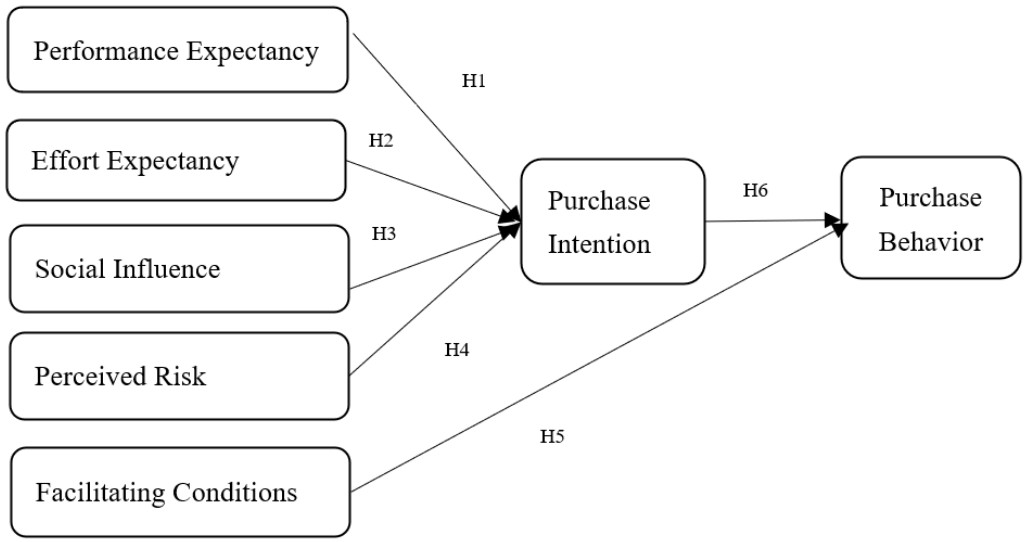

**Figure 1.** Research model.

### 3.1. Performance Expectancy

Performance expectancy refers to the practicability of the system, and emphasizes that the system can improve customer performance. That is, consumers with higher expectations of OCGB performance would be more willing to use it [13,19–21]. Mäntymäki and Salo (2013) [22] found that consumers' PE for group buying in virtual communities will positively affect their group buying intentions. Doan (2020) [23] confirmed that PE positively affects consumers' online shopping intentions. Nur and Panggabean (2021) [15] reported that PE significantly affects the behavioral intentions of online transactions using mobile payments. Dewi et al. (2019) [24] showed that PE is the main factor affecting online purchase intention. Sharifi Fard et al. (2016) [25] PE to be the main factor affecting Malaysian users' online purchase intention through social networking sites. Juaneda

Ayensa et al. (2016) [26] reported that PE is the key determinant of purchase intention. Thus, we propose the following hypothesis:

**H1:** *Performance expectancy has a positive effect on OCGB purchase intention.*

### 3.2. Effort Expectancy

Effort expectancy means that when people will be more willing to use information systems when they learn to use them. That is, when consumers use OCGB with a relatively simple, purchase process and a relatively easy to use platform system, they will be more willing to use it [19,21,27]. Many scholars pointed out that effort expectancy positively affect the willingness to use [13,28,29]. For example, Escobar-Rodríguez and Carvajal-Trujillo (2014) [30] proposed that the ordering process's clarity and ease of use would significantly affect consumers' willingness to purchase online tickets. In online shopping for fresh agricultural products, if the operation process is relatively simple and easy to master, it will increase consumers' enthusiasm for shopping [31]. An et al. (2016) [32] pointed out that EE positively affects the online shopping intention of agricultural products. Thus, we propose the following hypothesis:

**H2:** *Effort expectancy has a positive effect on OCGB purchase intention.*

### 3.3. Social Influence

Social influence refers to the influence of individual consciousness or social network on the interaction between people. In this research, it refers to the degree to which consumers feel the influence of the opinions of important people when using OCGB, such as family and friends [13,33]. Social influence plays an important role in the decision to use the system [34–36]. Oliveira et al. (2016) [37] found that social influence significantly affects the willingness to use mobile payment. Yang (2010) [38] found that social impact plays a key role in mobile shopping use intention when studying mobile shopping services in the United States. Doan (2020) [23] determined that social influence was the most significant factor influencing Vietnamese online shopping willingness. Wei et al. (2021) [39] found that social influence has positive effect on the young generation's behavioral intention to adopt mobile payment. Chen et al. (2021) [16] found that social influence significantly affects consumers' willingness to buy fresh food through e-commerce platforms. Thus, we propose the following hypothesis:

**H3:** *Social influence has a positive effect on OCGB purchase intention.*

### 3.4. Perceived Risk

Bauer (1967) [40] defined perceived risk as a combination of uncertainty and the severity of the results involved. The perceived risk in this research refers to the uncertainty of consumers' use of OCGB and the possibility of economic and psychological losses caused by wrong purchase decisions. Users are prone to generate perceived risks in shopping [41,42]. Bhukya and Singh (2015) [43] tested the dimensions of perceived risk and the effects of consumers' willingness to buy retailers' private brands. Liew and Falahat (2019) [44] pointed out that perceived risk is the main determinant of an online group purchase. Ariffin et al. (2018) [45] found that the five perceived risk factors significantly negatively affect consumers' online purchase intention. Samadi and Yaghoob-Nejadi (2009) [46] found that higher perceived risk led to lower willingness to purchase from the Internet in the future. The more consumers perceive the risk of online shopping products, the more reluctant they are to use online shopping [47–49]. Thus, we propose the following hypothesis:

**H4:** *Perceived risk has a positive effect on OCGB purchase intention.*

*3.5. Facilitating Conditions*

Facilitating conditions referto asset availability and support conditions when using the system. Facilitating conditions refer to the degree of objective conditions supporting consumers to use OCGB, such as knowledge, resources, technology, and equipment [13,21]. Yu et al. (2019) [50] found that facilitating conditions have a positive effect on the use behavior of agricultural e-commerce users. Palau-Saumell et al. (2019) [51] reported that facilitating conditions significantly influence users' utilization of mobile applications for restaurant searches. Rakhmawati and Rusydi (2020) [52] pointed out that facilitating conditions significantly affect consumers' willingness to use RFID services. Nur and Panggabean (2021) [15] determined that facilitating conditions significantly affected the behavioral intentions of online transactions using mobile payments. It is generally agreed that facilitating conditions are the most significant factor in determining consumer usage behavior [53–55]. Thus, we propose the following hypothesies:

**H5:** *Facilitating conditions have a positive effect on OCGB purchase behavior.*

*3.6. Purchase Intention*

Purchase intention refers to the subjective possibility that individuals are willing to use OCGB at a certain time or under certain circumstances [56–58]. Jamil and Mat (2011) [59] proposed that purchase intention positively affects actual online purchases. Zeng et al. (2016) [60] found that purchase intention significantly affects consumers' online agricultural product purchase behavior. Liu (2020) [61] discussed the influence mechanism of individual perception and social influence on group-buying behavior and showed that willingness to use significantly positively affects online group-buying behavior. Intention is considered to be the key factor of actual behavior [56,62–64]. Thus, we propose the following hypothesis:

**H6:** *Purchase intention has a positive effect on OCGB purchase behavior.*

## 4. Data and Methodology

An online survey was distributed in April and May 2022 to collect data for this research. The target population of this survey is residents who live in China and have experience in OCGB. A convenient sampling technique is adopted. The questionnaire invited members with online group purchase experience to support the survey by positing a hyperlink on the questionnaire star. All questions were based on the respondents' experience of shopping on OCGB platforms. Three hundred questionnaires were collected. However, 280 of the respondents are valid. Demographic data are shown in Table 1. I Informed consent of all participants was obtained and participation was voluntary.

The questionnaire design of this research, based on UTAUT model and perceived risk theory, had 28 measurement items. All items measuring the research structure are measured with a 5-point Likert scale with 1 meaning "highly disagreed", 2 "disagreed", 3 "average", 4 "agreed", and 5 "highly agreed". The questionnaire consists of three parts. In the first part, the purpose of the questionnaire is described. The second part is the 28 items of the online purchase intention influencing factors scale of community group purchase consumers. The third part contains the basic demographic characteristics, including gender, age, occupation, and educational background.

Table 1 sows that among the 280 valid questionnaires, 31% were males and 68.9% were female, with a ratio of about 1:2. In terms of age distribution, the majority are 18–25 years old, accounting for 68.5% of the total. In terms of educational background distribution, most of the participants were undergraduates, accounting for 67.1% of the total. Regarding occupation distribution, the majority were students, accounting for 62.5% of the total. In terms of the distribution of disposable income, because most of the respondents are students and have little labor income, and the monthly living expenses are their disposable income, disposable income below 2000 accounted for the largest proportion at 58.2%.

**Table 1.** Demographic details of the respondents (*n* = 280).

| Consumer Statistics | Options | Frequency | Percentage |
|---|---|---|---|
| Gender | male | 87 | 31% |
| | Female | 193 | 68.9% |
| Age | 18 and under | 8 | 2.8% |
| | 18–25 years old | 192 | 68.5% |
| | 25–35 years old | 48 | 17.1% |
| | 35–45 years old | 21 | 7.5% |
| | 45 years old and above | 11 | 3.9% |
| | Junior high school and below | 26 | 9.2% |
| | high school | 14 | 5% |
| Education | Specialist | 29 | 10.3% |
| | Undergraduate | 188 | 67.1% |
| | Master's degree and above | 23 | 8.2% |
| | student | 175 | 62.5% |
| | Personnel of agencies and enterprises | 25 | 8.9% |
| Profession | social worker | 49 | 17.5% |
| | individual practitioners | 10 | 3.5% |
| | other | 21 | 7.5% |
| | Below 2000 | 163 | 58.2% |
| Monthly disposable income | 2000–4000 | 79 | 28.2% |
| | 4000–6000 | 22 | 7.8% |
| | 6000–8000 | 8 | 2.8% |
| | Above 8000 | 8 | 2.8% |

## 5. Results

To solve this research problem, the analysis was conducted using SPSS 25.0 (IBM, Armonk, NY, USA). Exploratory factor analysis and reliability analysis were conducted to verify the reliability and validity of each variable. The correlation analysis of the data was conducted to determine how various variables are related. In addition, regression analysis was conducted to test the hypotheses.

### 5.1. Reliability and Validity Analysis

Validity analysis was conducted based on Eigenvalue 1 and factor loading 0.5, and reliability analysis was conducted based on Cronbach's alpha 0.6. The results of the analysis were shown in Table 2. Seven factors with an eigenvalue of 1 or more were derived. Factor loading of all factors was 0.5 or more. In addition, Cronbach's alpha coefficient was 0.6 or more, and thus, the reliability of the variable was also recognized.

**Table 2.** Analysis of reliability and validity of research variables.

| Variable | Item | Factor Loading | Eigenvalue | Total Variance Explained (%) | Cronbach's Alpha |
|---|---|---|---|---|---|
| Performance Expectancy (PE) | PE1 | 0.875 | | | |
| | PE2 | 0.869 | >1 | 76.607 | 0.898 |
| | PE3 | 0.878 | | | |
| | PE4 | 0.879 | | | |
| Effort Expectancy (EE) | EE1 | 0.831 | | | |
| | EE2 | 0.857 | >1 | 69.806 | 0.855 |
| | EE3 | 0.862 | | | |
| | EE4 | 0.790 | | | |
| Social Influence (SI) | SI1 | 0.857 | | | |
| | SI2 | 0.894 | >1 | 75.910 | 0.894 |
| | SI3 | 0.881 | | | |
| | SI4 | 0.852 | | | |

**Table 2.** *Cont.*

| Variable | Item | Factor Loading | Eigenvalue | Total Variance Explained (%) | Cronbach's Alpha |
|---|---|---|---|---|---|
| Facilitating conditions (FC) | FC1 | 0.852 | | | |
| | FC2 | 0.861 | >1 | 72.660 | 0.874 |
| | FC3 | 0.863 | | | |
| | FC4 | 0.832 | | | |
| Perceived Risk (PR) | PR1 | 0.840 | | | |
| | PR2 | 0.871 | >1 | 72.038 | 0.870 |
| | PR3 | 0.884 | | | |
| | PR4 | 0.797 | | | |
| Purchase Intention (UI) | PI1 | 0.863 | | | |
| | PI2 | 0.882 | >1 | 74.289 | 0.884 |
| | PI3 | 0.876 | | | |
| | PI4 | 0.826 | | | |
| Purchase Behavior (UB) | PB1 | 0.839 | | | |
| | PB2 | 0.880 | >1 | 69.134 | 0.848 |
| | PB3 | 0.795 | | | |
| | PB4 | 0.809 | | | |

*5.2. Correlation Analysis*

Correlation analysis is the premise and foundation of regression analysis. I This research used the Pearson correlation coefficient method to determine the correlation between variables. Table 3 shows the results of the variable correlation analysis. Table 4 shows that, at the significance level of 0.01, the correlation coefficient between performance expectancy and purchase intention is 0.628, effort expectancy and purchase intention is 0.633, social influence and purchase intention is 0.699, perceived risk and purchase intention is 0.389, the correlation coefficient between facilitating conditions and purchase behavior is 0.584, and the correlation coefficient between purchase intention and purchase behavior is 0.818. Therefore, there is a positive correlation between the variables.

**Table 3.** Correlation analysis.

| | 1 | 2 | 3 | 4 | 5 | 6 | 7 |
|---|---|---|---|---|---|---|---|
| 1. Performance Expectancy | | | | | | | |
| 2. Effort Expectancy | 0.601 ** | | | | | | |
| 3. Social Influence | 0.592 ** | 0.748 ** | | | | | |
| 4. Facilitating conditions | 0.597 ** | 0.672 ** | 0.766 ** | | | | |
| 5. Perceived Risk | 0.327 ** | 0.426 ** | 0.472 ** | 0.453 ** | | | |
| 6. Purchase intention | 0.628 ** | 0.633 ** | 0.699 ** | 0.666 ** | 0.398 ** | | |
| 7. Purchase behavior | 0.545 ** | 0.511 ** | 0.584 ** | 0.584 ** | 0.340 ** | 0.818 ** | |

** $p < 0.01$.

**Table 4.** Regression results for hypotheses.

| Dependent Variable | Purchase Intention | Purchase Behavior |
|---|---|---|
| Model | Model 1 | Model 2 |
| | coefficient | coefficient |
| Performance expectancy | 0.292 *** | |
| Effort expectancy | 0.136 * | |
| Social influence | 0.397 *** | |
| Perceived Risk | 0.058 | |
| Facilitating conditions | | 0.071 |

**Table 4.** *Cont.*

| Dependent Variable | Purchase Intention | Purchase Behavior |
|---|---|---|
| Purchase intention | | 0.771 *** |
| R square | 0.570 | 0.671 |
| Adjust R square | 0.563 | 0.669 |
| F | 90.975 *** | 282.995 *** |

\* $p < 0.05$, \*\*\* $p < 0.001$.

*5.3. Regression Analysis*

The hypotheses were tested using regression analysis. Table 4 shows the results of the regression analysis. Model 1 in Table 4 refers to the regression analysis results between performance expectancy, effect expectancy, social influence, and perceived risk and purchase intention. The results show that performance expectancy (coefficient = 0.292, $p < 0.001$), effort expectancy (coefficient = 0.136, $p < 0.05$), and social influence (coefficient = 0.397, $p < 0.001$) have a significant effect purchase intention. However, perceived risk (coefficient = 0.058, $p$ = n.s) does not affectpurchase intention. Therefore, hypotheses H1, H2, and H3 are supported, and H5 is not supported.

Model 2 in Table 4 refers to the regression analysis results between facilitating conditions, purchase intention, and purchase behavior. The results show that facilitating conditions (coefficient = 0.071, $p$ = n.s) do not affect purchase behavior. However, purchase intention (coefficient = 0.528, $p < 0.001$) significantly affects purchase behavior. Therefore, hypothesis H4 is not supported and H6 is supported. The hypotheses test results are shown in Table 5.

**Table 5.** Hypotheses test results.

| | Hypotheses | Findings |
|---|---|---|
| H1 | Performance expectancy has a positive effect on the online purchase intention of community group-purchase consumers. | Supported |
| H2 | Effort expectancy has a positive effect on the online purchase intention of community group-purchase consumers | Supported |
| H3 | Social influence has a positive effect on the online purchase intention of community group-purchase consumers | Supported |
| H4 | Perceived risk has a positive effect on the online purchase intention of community group-purchase consumers | Not Supported |
| H5 | Facilitating conditions has a positive effect on the online purchase behavior of community group-purchase consumers | Not Supported |
| H6 | Purchase intention has a positive effect on the online purchase behavior of community group-purchase consumers. | Supported |

**6. Discussion**

This paper focused on the purchasing intention and influencing factors of OCGB consumers. Consumers who have utilizedcommunity group-buying are taken as the survey objects; the UTAUT model and perceived risk theory are taken as the theoretical basis; performance expectancy, effort expectancy, social influence, facilitating conditions, and perceived risk are taken as independent variables, and purchase intention and purchase behavior are taken as dependent variables. This research establishes a model of OCGB consumers' online purchase intention and influencing factors and conducts an empirical analysis of the collected data. The collected data were analyzed using SPSS25.0, and the results are as follows.

*6.1. Performance Expectancy Has a Positive Effect on OCGB Purchase Intention*

From the results of empirical analysis, the regression coefficient between performance expectancy and purchase intention is 0.256, which indicates that the higher the perfor-

mance expectancy of consumers, the higher their purchase intention. The majority of the existing literature supports this view [13,65,66]. The results show that if consumers can buy goods more conveniently through OCGB, can buy cheap goods, and can shop anytime and anywhere, regardless of other factors, consumers' performance expectancy will be enhanced.

### 6.2. Effort Expectancy Has a Positive Effect on OCGB Purchase Intention

According to the results of empirical analysis, the correlation coefficient between effort expectancy and purchase intention is 0.633, and the regression coefficient is 0.130, indicating that the higher the effort expectancy of consumers, the higher their purchase intention. The majority of the existing literature supports this view [65,67]. Suppose the OCGB platform is simpler in operation, simpler in processes and links, and easier to master useful skills; in that case it will enhance consumers' expectations of effort, thus significantly enhancing their willingness to use OCGB.

### 6.3. Social Influence Has a Positive Effect on OCGB Purchase Intention

According to the results of empirical analysis, the regression coefficient between social influence and purchase intention is 0.398, which has a significant positive effect on purchase intention. The majority of the existing literature supports this view [36,68]. This result shows there is a "zero distance" contact between consumers in the information age, and product promotion information will spread rapidly on the Internet. OCGB will positively affect consumers when their friends, colleagues, and online comments approve or encourage it. As a result, consumers are more likely to try using the OCGB platform.

### 6.4. Perceived Risk Has No Impact on OCGB Purchase Intention

In this research, the regression coefficient between perceived risk and purchase intention is 0.052, but its significance level is 0.203. Thus, perceived risk has no significant effect on online purchase intention. The majority of the existing literature supports this view [43,69]. Perception of risk may be related to specific factors [70]. The perceived risk of frequent online shoppers is often low [67], which may be because the respondents in our research are experienced in online shopping, and hence, the perceived risk has little effect. At the same time, with the improvement of China's information security technology and its network legal mechanism, consumers have gradually taken a positive attitude towards the confidentiality and security of the OCGB platform, gradually weakening consumers' perceived risk.

### 6.5. Facilitating Conditions Have No Effect on OCGB Purchase Intention

This research found that the interface design of the community group-buying platform, Internet speed, quality and safety certification of products, and high-quality return and exchange services have no direct influence on the purchasing behavior of consumers. Tam et al. (2018) [19] showed that our interviewees are integrating smart phones into their daily life, and with the ideal conditions for using mobile applications, they no longer attach importance to convenience.

### 6.6. Purchase Intention Has a Positive Effect on OCGB Purchase Behavior

From the empirical analysis results, the regression coefficient between purchase intention and purchase behavior is 0.818, indicating that purchase intention significantly affects purchase behavior. The majority of the existing literature supports this finding [20,60]. Consumers' purchase behavior is directly affected by their purchase intention.

## 7. Conclusions

The effects of the COVID-19 pandemic has increased the potential of OCGB platform in retail [16]. Based on the UTAUT model and perceived risk theory, this research constructed a theoretical model of consumers' OCGB intention and behavior and tested the reliability

and validity of valid sample data. Hypotheses testing and path analysis of the model were also conducted that the results demonstrated that consumers' willingness to use OCGB is significantly influenced by performance expectancy, effort expectancy, and social influence. Facilitating conditions and perceived risk have no significant effect. Therefore, we put forward suggestions on the construction and operation of the OCGB platform.

One of the reasons consumers use a community group-buying platform is that it improves shopping efficiency of consumers. Community group-buying platform helps consumers to buy the goods they want by moving their fingers on mobile devices without leaving home. Therefore, a community group-buying platform can collect consumers' purchase needs and make mass purchases, which has the advantage of bulk purchases. Consumers no longer require "shopping around" to buy high-quality and cheap goods without leaving their homes. Similarly, designing a more simple and easy-to-learn platform operating system with the commodity information on the platform in a popular language can enable more people to learn to use it in a short time and be familiar with the operation process will enable more people to participate in the OCGB platform, thereby facilitating more passenger flow.

Furthermore, from the perspective of integrated marketing communication strategy, the target audience is the immediate consumers, potential consumers, and other relevant personnel. Hence, the OCGB platform needs to understand the needs of consumers when formulating marketing strategies to design incentives, and highlight the characteristics of the platform to transmit information to consumers, stimulate consumers' desire to buy, and provide timely guidance to consumers to produce purchasing behavior. Finally, consumers' intentions to make online purchases are significantly influenced by social influence. From the perspective of consumers, there are two main sources of information for consumers. One is consumers' sources, including their families, friends, neighbors, acquaintances, and other people who have contact with them. The second is public sources, including reports from mass media, social media, and third-party organizations. For the OCGB platform, what can be changed is to make good use of public relations, show consumers a good platform image, and be recognized and accepted by consumers, so that consumers can develop the habit of using the OCGB platform continuously.

## 8. Limitations

This research conducted a comprehensive empirical analysis on the influencing factors of the online purchase intention of OCGB consumers. Although it provides a reference for the OCGB platform's sustainable development, it the following limitations. First, based on retaining the four main variables of the UTAUT model, this research adds perceived risk as a new variable. Future research can add different structures to expand applicability, considering consumers' personality characteristics or other influencing factors, such as individual innovation and trust. The interviewees in this research come from all over China. Although they are representative, different regions have different levels of economic development and consumption habits. Therefore, the regional differences affect consumers' purchase intention and behavior in community group buying. The results of this research depend on the quantitative research conducted through the questionnaire survey. Future research can better understand consumers' purchase intentions through qualitative methods or a combination of qualitative and quantitative methods.

**Author Contributions:** Conceptualization, H.H.; Methodology, M.W.M.; Writing—original draft, M.Z. All authors have read and agreed to the published version of the manuscript.

**Funding:** This research received no external funding.

**Institutional Review Board Statement:** Not applicable.

**Informed Consent Statement:** Not applicable.

**Data Availability Statement:** Data can be made available on request.

**Conflicts of Interest:** The authors declare no conflict of interest.

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
