# Peer review of "Exploring the Consumers’ Purchase Intention on Online Community Group Buying Platform during Pandemic"

_sustainability, doi:10.3390/su15032433_

Round 1
Reviewer 1 Report
The research on the topic of the consumer behavior during the pandemic is important and great to discover. The considerations presented in the article are very current and interesting.
But as for details I would like to ask authors to correct before printing, then:
1. In the title it is worth adding information that it concerns China.
2. In the abstract, the context of the research should be better explained, so as the aim. A real aim of the paper is not explained. Authors explain what they explore but the aim of the research is not stated and therefore cannot be understood.
3. There is no information about shopping behavior in China or only in a given province until the pandemic.
4. What is the share of "Community group buying" in all online and offline purchases?
5. Literature review is not enough to support the literature gap considered in this study. The introduction must be substantially integrated through a more exhaustive literature review, with the aim of highlighting the literature gaps and justifying the research objectives.
6. The Figure 1 needs to be corrected, some elements have gone apart.
- How the sample was designed and approached to collect the data whether the sample is representative?
Reviewer 2 Report
Dear Authors
Thank you for submitting your manuscript, "Exploring the influencing factors of consumers' intention to use community group buying in a pandemic" for review. Overall, the study presents a thorough examination of the factors that influence online purchase intention in the context of community group buying. The use of the UTAUT model and perceived risk theory provides a solid theoretical foundation for the research. However, despite these strengths, the manuscript does have some weaknesses that need to be addressed before it can be considered for publication. Please address the following comments in your revised manuscript:
Please revise the abstract in the light of following comments
Abstract
-The abstract does not clearly state the research question or main objectives of the study.
-It would be helpful to provide more background information on community group buying and its importance in China.
-The abstract does not mention the data sources or sampling methods used in the study, which makes it difficult to evaluate the validity of the results.
-The abstract does not provide any details on the statistical analyses conducted or the results obtained, making it difficult to assess the contribution of the study.
-The conclusion of the study is not summarized in the abstract, making it difficult to understand the main takeaways of the research.
Introduction
The introduction does not provide a clear overview of the research question or the main objectives of the study. It would be helpful for the authors to clearly define the research question and provide a more comprehensive overview of the motivations behind the study.
Theoretical Background
The theoretical background section could benefit from a more detailed and comprehensive review of the literature on the UTAUT model and perceived risk theory. The authors should also clearly explain the relevance of these theories to the research question at hand.
I believe there are also some areas where the literature review could be further strengthened. Specifically, I believe it would be beneficial to the study to include more recent research on consumer purchase intentions in the context of the COVID-19 pandemic. In addition, it would be helpful to provide more context and background on the relevant concepts described in the study, such as the UTAUT model and perceived risk theory.
I recommend that you consider citing the following studies:
https://doi.org/10.3390/ijerph18031175https://doi.org/10.1111/cjag.12184
Research Model and Hypotheses
The research model and hypotheses section could be more clearly presented. It would be helpful if the authors could provide more detailed explanations of the variables included in the model, as well as the specific hypotheses being tested. It would also be helpful if the authors could provide more justification for the inclusion of certain variables in the model.
Additionally, it would be helpful if the authors could provide more information on the sample size and sampling techniques used in the study. This information is important for evaluating the generalizability of the results.
Research Methodology
The research methodology section does not provide enough information on the sample size and sampling techniques used in the study. It would be helpful to include more details on the sampling process, such as the inclusion and exclusion criteria for the sample, and the method used to ensure the representativeness of the sample.
The use of Likert scale for measuring the variables in the questionnaire should be more clearly explained, including the assumptions underlying this method and how it was applied to the data.
It would be helpful to provide more information on the data collection process, such as the mode of administration and the response rate. The effective recovery rate of 84.8% does not provide sufficient information on the overall response rate.
Results
The Results section appears to present the results of the analysis conducted on the data collected for the study. However, there are several issues with the way the results are presented that make it difficult for the reader to understand the findings of the study.
One issue is the lack of clear explanation of the statistical methods used. While the authors mention the use of Cronbach's alpha and exploratory factor analysis, they do not provide any further information on these methods or how they were applied in the study. This makes it difficult for readers to understand the significance of the results and how they were obtained.
Another issue is the lack of clear and concise interpretation of the results. The authors present a large number of tables and figures without providing any context or explanation for the results they present. This makes it difficult for readers to understand the findings of the study and how they relate to the research questions.
Finally, the results are not presented in a logical and coherent manner, making it difficult for readers to follow the flow of the analysis. For example, the authors present the results of the KMO test and Bartlett sphericity test before discussing the reliability of the questionnaire, which may be confusing for readers. Additionally, the authors do not provide any conclusions or implications based on the results of the analysis, making it difficult for readers to understand the overall significance of the study.
Conclusion
The conclusions of the study appear to be based on the results of the data analysis, but it is not clear how these results were interpreted or how they support the stated conclusions. Additionally, the discussion and implications section does not provide any further insight into the study or its findings. It simply states that the research is of "certain significance" and that future research could examine the operation mechanism of community group buying from multiple perspectives. It does not provide any in-depth analysis or discussion of the implications of the study or its results. Without more context or analysis, it is difficult to fully evaluate the significance or value of the research.
Reviewer 3 Report
Exploring the influencing factors of consumers' intention to use 1 community group buying in a Pandemic - Title
Introduction
The author must clearly explain the purpose of the research in the introduction.
The author should show why he deals with a certain topic,
Why is it important for a wide range of readers, scientists, researchers...
Why is the research significant and what is the contribution of your research
What gap in the literature does your research fill?
If other authors have dealt with the topic, why was it important for you to investigate (for your country and other countries).
Why the UTAUT method is significant
Literature view
The author should consult many more literature sources. It should present the conclusions of the authors who dealt with that topic. The second group of views refers to the application of the UTAUT model.
The literature should be cited in accordance with the purpose of the research and the set hypotheses.
Research model and hypotheses
It is not clear why Research model and hypotheses and Research methodology were separated. I suggest that it be merged and supplemented significantly.
Describe the questionnaire, in detail; questions, structure, respondents, sample, how you selected the respondents, representativeness of the sample, according to which research was created...
Hypotheses are derived based on a detailed presentation of the conclusions of other authors. Therefore, for each hypothesis you should list at least 10 references and show how you came to it and why you put it forward. What are you going to show? The model was certainly created on the basis of the analysis of some study, it must be mentioned and elaborated further.
Describe the UTAUT model, why it was applied, when it is applied, what are its characteristics. Who has used it before and why. What are the criteria and stages of the analysis.
Research methodology
Rename the section to research results.
I would put the discussion before the conclusion and when you introduce all the literature, discuss their results and yours.

Round 2
Reviewer 1 Report
Thank you for the corrections.
Author Response
Thank you for submitting a much-improved manuscript. Could you please send the paper through additional round of polishing. Also, review the tables for unnecessary information. For instance, how do "Model 1" and "Model 2" headings help the reader. Also, why use beta? You can simply label the column as "coefficient."
Response:
Thank you for your comments. We have explained model1 and model 2 on page 15 and 16 respectively,the beta has been changed to coefficient. Please see changes in blue.
Reviewer 2 Report
Authors have made satisfactory revisions in response to the original review. The paper can be accepted now.
Author Response
When summarizing findings vis-a-vis hypothesis, the columns could be labeled "Hypotheses" and "Findings."
Response:
Thank you for your comments. These columns have been labeled "Hypotheses" and "Findings." Please see changes in blue.
Reviewer 3 Report
The author has made the requested corrections.
Author Response
There is some choppiness in the language and some word choices are not ideal for a technical/academic article.
Response:
Thank you for your comments. We have adjusted the sentences and revised the words. Please see changes in blue.